# Development of Ozone Reactivity Scales for Volatile Organic Compounds in a Chinese Megacity

Yingnan Zhang[1], Likun Xue[1,2*], William P. L. Carter[3], Chenglei Pei[4,5,6,7], Tianshu Chen[1], Jiangshan Mu[1], Yujun Wang[6], Qingzhu Zhang[1], Wenxing Wang[1]

[1]Environment Research Institute, Shandong University, Ji'nan, Shandong, China
[2]Collaborative innovation Center for climate Change, Jiangsu Province, Nanjing, China
[3]Center for Environmental Research and Technology, University of California, Riverside, Riverside CA, USA
[4]State Key Laboratory of Organic Geochemistry and Guangdong Key Laboratory of Environmental Protection and Resources Utilization, Guangzhou Institute of Geochemistry, Chinese Academy of Sciences, Guangzhou, China
[5]University of Chinese Academy of Sciences, Beijing, China
[6]Guangzhou Ecological and Environmental Monitoring Center of Guangdong Province, Guangzhou, China
[7]Guangdong Provincial Observation and Research Station for Climate Environment and Air Quality Change in the Pearl River Estuary, Guangzhou, China

*Correspondence to*: Likun Xue (xuelikun@sdu.edu.cn)

**Abstract.** We developed incremental reactivity (IR) scales for 116 volatile organic compounds (VOCs) in a Chinese megacity (Guangzhou) and elucidated their application in calculating the ozone ($O_3$) formation potential (OFP) in China. Two sets of model inputs (emission-based and observation-based) were designed to localize the IR scales in Guangzhou using the Master Chemical Mechanism (MCM) box model, and were also compared with those of the U.S. The two inputs differed in how primary pollutant inputs in the model were derived, with one based on emission data and the other based on observed pollutant levels, but the maximum incremental reactivity (MIR) scales derived from them were fairly similar. The IR scales showed a strong dependence on the chemical mechanism (MCM vs. Statewide Air Pollution Research Center), and a higher consistency was found in IR scales between China and the U.S. using a similar chemical mechanism. With a given chemical mechanism, the MIR scale for most VOCs showed a relatively small dependence on environmental conditions. However, when the NOx availability decreased, the IR scales became more sensitive to environmental conditions and the discrepancy between the IR scales obtained from emission-based and observation-based inputs increased, thereby implying the necessity to localize IR scales over mixed-limited or NOx-limited areas. This study provides recommendations for the application of IR scales, which has great significance for VOC control in China and other countries suffering from serious $O_3$ air pollution.

## 1 Introduction

Tropospheric ozone ($O_3$) is an important component of photochemical air pollution and exerts great effects on atmospheric chemistry, climate change, and human and vegetation health (Agathokleous et al., 2020; Fleming et al, 2018; IPCC, 2013; Lefohn et al., 2018; Lelieveld et al., 2008; Mills et al., 2018; Monks et al., 2015). Surface $O_3$ pollution has become a major air quality issue in China in recent decades (Wang et al., 2017; and references therein). On a national scale, the $O_3$ pollution

situation has worsened despite significant improvements in other routinely monitored pollutants (e.g., $SO_2$, $NO_2$, CO, $PM_{2.5}$,

and $PM_{10}$) (Chen et al., 2020; Li et al., 2019a; Liu and Wang, 2020; Sun et al., 2016; Wang et al., 2009; Wang et al., 2019; Xu et al., 2020; Zheng et al., 2018). Alleviating $O_3$ pollution is the key to evaluating air quality control effectiveness during the Blue Sky Protection Campaign, and is also the focus of air quality management in the 14th Five-Year Plan (2021-2025). Considering the significance of volatile organic compounds (VOCs) to $O_3$ formation, growing importance has been attached to the VOC management. The $O_3$ formation potential (OFP) scale has been extensively used to quantify the relative effects

of individual VOCs on $O_3$ formation and to aid in the development of cost-effective control strategies (Carter and Atkinson, 1989; Chang and Rudy, 1990).

For a given VOC or VOC mixture, the OFP is determined by two factors, namely kinetic reactivity (KR) and mechanistic reactivity (MR) (Carter, 1994a; Carter and Atkinson, 1989; Venecek et al., 2018). KR refers to the fraction of the given VOC (mixture) being reacted in a scenario, and MR refers to the $O_3$ change caused by the reaction of the amount of the given

VOC (mixture). To further quantify the OFP, Carter and Atkinson (1989) and Carter (1994a) defined two operational concepts of incremental reactivity (IR) and relative reactivity (RR). IR is defined as the grams of $O_3$ change per gram of VOCs added to a base mixture, and RR is calculated as the IR of the target VOC divided by the IR of a reference VOC. Generally, IR is used for absolute OFP quantification, and RR is used for relative OFP (ROFP) quantification (Chang and Rudy, 1990). The original IR scale was developed using a chemical box model built upon a detailed chemical mechanism

(Statewide Air Pollution Research Center; SAPRC), and the adopted model scenarios were representative of pollution conditions in 39 urban areas of the U.S. (Carter, 1994a). The maximum incremental reactivity (MIR) scale developed under high NOx conditions (i.e., VOC-limited $O_3$ formation regime) has become the most popular scale for quantifying the OFP in scientific studies related to $O_3$ mitigation programs. The application of such calculations of the OFP scale is restricted to areas or episodes in which the $O_3$ formation regime is VOC-limited or at least mixed-limited by VOCs and NOx.

Over the past two decades, two major concerns have been raised regarding the application of IR and OFP scales. One originated from IR localization. The original IR values have been updated in several studies owing to chemical mechanism (MR) updates and environmental changes in the U.S. (Bergin et al., 1998; Carter, 2009; Carter, 1994b, 2010; Venecek et al., 2018). The most commonly used version of the IR scale was provided by Carter et al. (2010) and Venecek et al. (2018). Considering the different atmospheric conditions (e.g., VOC compositions and relative availability of VOCs and NOx)

between the U.S. and other countries (Kurokawa and Ohara, 2020; Li et al., 2019b; Venecek et al., 2018), it is unknown whether the IR scale constructed for the U.S. can be mechanically used for reactive VOC identification in other countries. Another concern is related to the role that observational data and emission data should play in establishing the environmental conditions that need to be represented in the IR and OFP calculations. Initially, IR values were determined using emission data and applied to quantify the OFP by multiplying the emission quantity or emission rate (Carter, 1994a; Li et al., 2019b;

McNair et al., 1994; Mo et al., 2020). In contrast, most studies in China (e.g., Cai et al., 2010; Hong et al., 2019; and Hui et al., 2020) multiplied the MIR values by the observed VOC concentrations to quantify the OFP. The differences in the IR

scales obtained from emission and observational data and how to rationally use observational data to determine OFP scales require further examination. Addressing these two concerns has great significance for $O_3$ control in China and other countries suffering from serious $O_3$ pollution.

In this study, we designed two sets of model inputs and adopted the Master Chemical Mechanism version 3.3.1 (MCMv3.3.1) box model to localize the IR scales in urban Guangzhou, a megacity of southern China. The two model inputs differed in how the primary pollutant inputs in the model for Guangzhou were derived, with one based on emission data and the other based on observed pollutant levels, whereas the other inputs were the same. The results obtained were used to elucidate how to apply IR and OFP scales. In this paper, we first present and compare the localized emission-based and observation-based

IR scales in urban Guangzhou. We then provide comparisons between the localized IR scales and U.S. IR scales and identify key factors affecting the IR scales of VOCs. Finally, we provide recommendations for the application of IR and OFP scales to aid in VOC control in Chinese cities.

## 2 Materials and methods

### 2.1 Typical urban pollution scenarios

Guangzhou is the regional center of the Pearl River Delta (PRD), one of the most developed regions in China (Figure S1). In 2019, Guangzhou covered 7434 $km^2$ and the urbanization rate was 86 %. The area is home to over 15 million people, and most of the residents live in urbanized districts. Two-year data from 1 January 2018 to 31 December 2019 used for IR localization were obtained from an urban site in Guangzhou (Figure S1). Detailed information on the sampling site, measurement techniques, and quality assurance/control procedures are provided in the SI.

The 1-day observation-based model inputs were median diurnal profiles of 67 $O_3$ episode days (Figure S2), in which the maximum daily 8-hour average $O_3$ mixing ratio exceeded the Chinese National Ambient Air Quality Standard, i.e., 75 ppbv (Class II). During the 67 $O_3$ episodes, the average amplitude (defined as the maximum minus the minimum) of the diurnal $O_3$ cycle was $104 \pm 23$ ppbv. The large amplitude indicated the significant influence of intense photochemical $O_3$ production during daytime as well as impact of NO titration and/or dry deposition during nighttime with shallow nocturnal boundary

layer. The weather conditions presented a high temperature ($27.7 \pm 3.7$ °C) and moderate relative humidity (RH; $49 \pm 14$ %). These weather conditions are favorable for $O_3$ formation. The $O_3$ precursors NOx and VOCs exhibited a morning peak and another evening peak in the rush hours, and the observed average NOx and VOC mixing ratios (38.05 and 39.16 ppbv, respectively; Table S1) were comparable to those in other megacities such as Beijing and Shanghai (Li et al., 2015; Liu et al., 2019; Sun et al., 2018). Within the observed VOC compositions (Figure 1a), alkanes made the largest contribution (59 %),

followed by aromatics (20 %), alkenes (15 %), and alkynes (6 %). The average VOC/NOx ratio was $0.99 \pm 0.46$ ppbv $ppbv^{-1}$. The diurnal variation in primary pollutants and high NOx concentrations were representative of urban pollution conditions greatly affected by fresh traffic emissions (with intense NOx emissions).

## 2.2 Chemical box model Setup

A chemical box model was adopted for the IR calculation. The model was run based on the platform of F0AM (Framework
for 0-D Atmospheric Modeling) (Wolfe et al., 2016), and built based on the near-explicit chemical mechanism of
MCMv3.3.1 (http://mcm.york.ac.uk/), which describes the degradation pathways of 143 primary VOCs in detail (Jenkin et
al., 2003; Saunders et al., 2003). In addition to state-of-the-art chemistry, the model also incorporated physical processes,
including solar radiation, diurnal evolution of the planetary boundary layer (PBL), dry deposition, and dilution with
background air. A detailed description of the model configuration is provided in previous studies (Xue et al., 2014; Xue et al.,
2013) and is also documented in the SI.

We designed two 1-day model inputs to compare the observation-based and emission-based IR scales (Table S2). With
observation-based inputs, the model was constrained by the median diurnal profiles of NO, $NO_2$, VOCs (39 compounds),
$SO_2$, and CO concentrations observed in the 67 selected $O_3$ episodes. While the NO and $NO_2$ inputs into the model were
determined by observational data with observation-based inputs, their evolution over time was determined by the chemistry
that was affected by the VOC-involved reactions. With emission-based inputs, the model read the median diurnal profiles of
the emission rates of NO, $NO_2$, VOCs (116 compounds), $SO_2$, and CO (see the SI for detailed methods to calculate the
emission rate) and was initialized using observational data at 06:00 local time (LT; the initial time of model integration)
(Table S3). The VOC composition factions calculated from the observed data were not the same as those calculated from the
emission data because observational data were only available for 39 compounds (Figure 1). With emission-based inputs, the
initial concentrations of VOCs without available observational data were set as 0.10 ppbv, except for those of formaldehyde
(0.50 ppbv) and acetaldehyde (0.30 ppbv). Such treatment of VOC initialization would inevitably cause uncertainty to the
obtained RRs. Sensitivity tests were conducted to evaluate the potential uncertainty, and the results are documented in Table
S4. The RRs (i.e., IR/Ethene in Table S4: IR value of a given VOC divided by the IR value of ethene) obtained from
sensitivity tests exhibited good correlations with those obtained from emission-based base case inputs ($R^2$ ranged from 0.98-
1.00), but minor discrepancy in RR magnitudes existed (reduced major axis (RMA) slope: 0.86-1.03). More high-quality
long-term observational data covering a variety of VOCs are highly needed for IR calculation.

In addition to the above differences, the two model inputs shared a common setup. Specifically, the model was constrained
by the median diurnal profiles of meteorological parameters, including temperature, RH, and pressure obtained from the 67
selected $O_3$ episodes; the HONO input was assumed to be equal to 2 % NOx input (Qin et al., 2009); radicals such as OH
and $HO_2$ were initialized according to 2-day pre-run results; and $O_3$ was initialized using the observed data at 06:00 LT, and
then its chemistry and concentrations were simulated with the constraints of other relevant species in the following
integration (Figure S3). For both model inputs, the model was run with 06:00 LT as the initial time, and the integration lasted
for 10 hours with a step of 1 hour. A series of sensitivity experiments (emission-based model inputs) with changing
environmental conditions were conducted to identify possible reasons for the IR discrepancy between Guangzhou and the
U.S.

## 2.3 Calculation of ozone reactivity scales for VOCs

The IR scales under three specified NOx conditions and base NOx conditions were localized in Guangzhou considering their dependence on NOx availability. Two major procedures, including NOx-adjusted runs and VOC-added runs, were used to calculate the IR scales. The detailed methods were illustrated in previous studies (Carter, 1994a, b). The exact NOx inputs for the three specified NOx conditions were determined according to the effect of a 1 % increase in VOC or NOx inputs on the model-simulated peak $O_3$ mixing ratio with varying scaling factors of base NOx inputs (Figure 2). The MIR scenario represents relatively high NOx conditions in which VOCs yield the highest IR. The maximum $O_3$ reactivity (MOR) scenario represents median NOx conditions, which are optimal for $O_3$ formation. The equal benefit incremental reactivity (EBIR) scenario represents relatively low NOx conditions in which VOC and NOx reduction exert equal effects on $O_3$ formation. When ambient NOx concentrations are lower than NOx conditions under EBIR scenarios, $O_3$ formation becomes primarily sensitive to NOx, and IRs for VOCs are less relevant to $O_3$ pollution. With observation-based inputs, the exact NOx inputs for the MIR, MOR, and EBIR scenarios were $1.15 \times [NOx]_{BASE}$, $0.60 \times [NOx]_{BASE}$, and $0.28 \times [NOx]_{BASE}$, respectively. $[NOx]_{BASE}$ represents the NOx inputs under base scenarios that were directly derived from observation or emission data, without any adjustment based on reactivity results. With emission-based inputs, the exact NOx inputs for the MIR, MOR, and EBIR scenarios were $1.03 \times [NOx]_{BASE}$, $0.66 \times [NOx]_{BASE}$, and $0.45 \times [NOx]_{BASE}$, respectively. Consistent with the results of previous studies (Tan et al., 2019; Xue et al., 2014), the NOx-adjusted results of both inputs indicated that the $O_3$ formation occurred in a VOC-limited regime in urban Guangzhou and the MIR scale is appropriate for urban Guangzhou conditions. This is probably also the case for other Chinese megacities (Ou et al., 2016; Xue et al., 2014). Identifying to which scale the base NOx condition has a good approximation is a prerequisite for determining the appropriate reactivity scale to apply.

With prescribed NOx inputs for a specified NOx condition, the base run and a series of VOC-added runs were simultaneously performed. In the VOC-added runs, a small amount of the target VOC was added while other inputs were kept unchanged relative to the base run. The criterion used to determine the addition amount of the target VOC was that the model-simulated peak $O_3$ change between the base run and VOC-added run should exhibit a near-linear relationship against the amount added (Carter, 1994a, b). To make a more precise determination, we changed the original method by increasing the target VOC input in the VOC-added runs from small to large folds (1–20 fold with a bin precision of 0.01), and fitted the obtained derivatives (with a precision of 0.01) of the model-simulated peak $O_3$ change relative to the target VOC change to a linear regression line to locate the point of interest. In this procedure, there was a difference in the treatment of the target VOC input between the emission-based and observation-based inputs. With emission-based inputs, the target VOC input was composed of the initial concentration and emission rate throughout the scenarios during both the base run and VOC-added run, and both the initial and emitted VOCs were increased from small to large folds in the series of VOC-added runs. The base run only needed to be performed once and used for the IR calculation for all of the individual VOCs. In contrast, with

the observation-based inputs, the target VOC was initialized but not constrained in the following integration, and only the initial target VOC was increased from small to large folds in the series of VOC-added runs. For the IR calculation of each VOC, a base run needed to be performed.

With both inputs, the amount of the target VOC input was not directly quantified, but calculated by adding an inert tracer species "TRACE" to be emitted and initially present in the same amount as that of the target VOC (TRACE was only initially present in the same amount as that of the target VOC with observation-based inputs as there were no emission data). The final calculated concentration of TRACE was used as a measure of the target VOC input (Carter, 1994b). The difference in the target VOC amount between the VOC-added run input and base run input was the small added amount. This calculation avoided complexities introduced by irrelevant physical factors such as the varying PBL height. The IR value of the target VOC was calculated as follows:

$$IR_i = \frac{\triangle O_3}{\triangle VOC_i} \qquad (1)$$

where $IR_i$ represents the IR of $VOC_i$, $\triangle O_3$ is the mass of additional $O_3$ formed, and $\triangle VOC_i$ is the amount of $VOC_i$ added to the scenario. IR values for 116 VOCs (39 VOCs) were calculated with emission-based (observation-based) inputs. Because 39 VOC species were the same within the two inputs, the IR scales for these common VOC species were compared, as discussed in Section 3.1. This is also why the IR scales for 111 (79) common VOC species are compared between Guangzhou and the U.S. (California) scenario used by Carter et al. (2010) (Derwent et al., 2010) in Section 3.2.

## 3 Results and discussion

### 3.1 Localized ozone reactivity scales for VOCs in Guangzhou

The IR scales for 116 VOCs under the MIR (MIR-Guangzhou), MOR (MOR-Guangzhou), EBIR (EBIR-Guangzhou), and base scenarios were localized (Figure 3 and Table 1; the results refer to emission-based IR scales unless stated). Under all three specified NOx conditions, alkenes were the most reactive VOC group (IR ranges of 2.46–14.11, 1.08–5.17, and 0.03–2.52 g g$^{-1}$ under the MIR, MOR, and EBIR scenarios, respectively). This was reasonable considering their large rate coefficients of reactions with OH radicals (Calvert et al., 2015; McGillen et al., 2020). Within the alkenes, *trans/cis*-2-butene was one of the most reactive compounds. The other reactive VOC groups were aromatics (0.38–9.64, -0.04–3.95, and -0.43–2.10 g g$^{-1}$ under the MIR, MOR, and EBIR scenarios, respectively) and oxygenated VOCs (OVOCs; -1.19–20.42, -1.36–7.30, and -1.65–3.42 g g$^{-1}$ under the MIR, MOR, and EBIR scenarios, respectively). The IR scales of both VOC groups exhibited a wide distribution and relied heavily on the NOx conditions. Within the aromatics, trimethylbenzene isomers were among the most reactive compounds. Within the OVOCs, aldehydes served as the most reactive sub-group, followed by ketones, alcohols, and other oxygenates. Biacetyl was the most reactive compound not only within OVOCs, but also within all of the defined VOCs. Consistent with the results of Carter et al. (2010), the IR values of benzaldehyde and its homologues were

negative because these compounds are strong radical and NOx sinks. A high proportion of compounds within the alkane (-0.10–1.29, -0.03–0.81, and -0.16–0.66 g g$^{-1}$ under the MIR, MOR, and EBIR scenarios, respectively), alkyne (0.20, 0.12, and 0.07 g g$^{-1}$ under the MIR, MOR, and EBIR scenarios, respectively), and halocarbon (0.002–3.03, 0.002–1.60, and 0.001–0.94 g g$^{-1}$ under the MIR, MOR, and EBIR scenarios, respectively) classes were unreactive or less reactive; therefore, these compounds are potential substitutes for more reactive compounds. Because the base NOx condition of urban Guangzhou was similar to the NOx condition of the MIR scenario, a high consistency was found between the base IR scale and the MIR scale (reduced major axis (RMA; Leduc, 1987) slope: 0.99; $R^2$ = 1), thereby indicating that O$_3$ formation occurred in a strongly VOC-limited regime and the MIR scale is the most appropriate for application in urban Guangzhou.

Despite the similarities, the IR scales showed significant dependence on the NOx availability, the degree of which varied among compounds (Figure 3). With the decrease in NOx availability, NOx became the key precursor to limit O$_3$ formation, and the IR magnitudes rapidly decreased. Compared with those in the MIR scenarios, the average IR values of VOCs significantly decreased under the MOR (by -61 %) and EBIR scenarios (by -82 %). Owing to the varying sensitivity to NOx availability, the relative ranks of several compounds also changed greatly. For example, the ranks of aldehydes (propionaldehyde, glyoxal, and butanal; Figure 3), which behave as significant radical sources (Zhang et al., 2019), decreased much faster under low NOx conditions, in which O$_3$ formation was limited more by the NOx availability than by the radical levels. The ranks of phenol and its homologues also presented marked deviations from the least square linear regression line. This was because the degradation products of phenol and its homologues behave as potent NOx sinks (i.e., forming HNO$_3$), which further lowers the NOx availability and reduces the IR of other VOCs. Overall, NOx availability is a dominant factor affecting both the IR ranks and magnitudes.

We also compared the MIR, MOR, and EBIR scales for 39 common VOC species obtained from both the emission-based and observation-based inputs (Figure 4). For most VOCs, a fairly good consistency of IR scales was found, as indicated by the modest to strong $R^2$ (0.95–0.98). The IR ranks of four VOCs (1-butene, ethyl benzene, $n$-dodecane, and styrene) differed significantly between the two inputs. These four compounds were markedly sensitive to environmental conditions, and the IR for these compounds reflected a balance between the positive effects on O$_3$ formation and the negative effects caused by radical inhibition. Although not mathematically significant ($p$ = 0.47, 0.25, and 0.19 under the MIR, MOR, and EBIR scenarios, respectively; tested by one-way analysis of variance method), the discrepancy of the IR magnitudes between the emission-based and observation-based inputs was non-negligible, and the discrepancy increased as the NOx availability decreased. Quantitatively, the average MIR values obtained from the observation-based inputs were lower (by -17 %) than those obtained by the emission-based inputs, but the average MOR and EBIR absolute values were higher (by 35 % and 49 %, respectively). Because both inputs shared similar physical configurations within the model, the discrepancy could have been related to the different chemical environmental conditions (such as VOC compositions) among the two inputs. Possible reasons for the discrepancy are discussed in Section 3.2. Overall, the comparison results between the two inputs

suggest that a general MIR scale obtained from either emission-based or observation-based inputs can be used for OFP calculations in most VOC-limited areas, but not in mixed-limited or NOx-limited areas.

## 3.2 Comparison with ozone reactivity scales for VOCs in U.S.

    MIR and MOR are the most appropriate for VOC-limited and mixed-limited conditions. Therefore, we focused on the comparison of these two scales between Guangzhou and a U.S. scenario (Figures 5 and 6). The MIR and MOR scales

obtained from Carter et al. (2010) were chosen to represent the U.S. scenarios and are referred to as MIR-Carter 2010 and MOR-Carter 2010 hereafter. MIR-Carter 2010 was the most popularly used version in previous Chinese studies. Both MIR-Carter 2010 and localized IR values in Guangzhou were calculated using chemical box models, but with different chemical mechanisms (SAPRC-07 vs. MCM v3.3.1). As shown in Figure 5, a non-negligible discrepancy was found for both the MIR and MOR scales ($p$ = 0.06 and 0.10, respectively) between Guangzhou and the U.S. scenarios, and the localized MIR and

MOR values in Guangzhou were generally lower (by -13 % and -3 %, respectively). Of all of the major VOC groups, the IR scales of the alkene class showed the lowest $R^2$ (MIR: 0.48; MOR: 0.42) by the least square regression, whereas those of the aromatic class showed the highest $R^2$ (MIR: 0.92; MOR: 0.94).

    To determine whether the discrepancy was caused by different environmental conditions between China and the U.S. or by different chemical mechanisms, we chose the MIR/Ethene scale developed using MCM v3.1 for further comparison, which

is referred to as MIR/Ethene-CA-MCM (Derwent et al., 2010). The MIR/Ethene scale was constructed for California conditions (a subset of the U.S. scenarios used by Carter (1994a, 2010)), but there was only a minor difference between the reactivity scales obtained from California scenarios and the average of all the U.S. scenarios using the SAPRC-07 mechanism (Figure S4), thereby indicating that the California scenarios can represent the U.S. reasonably well. With the same chemical mechanism (MCM), the MIR/Ethene scale resulted in higher consistency between China and California than

did the U.S. (Figure 6). The two calculations used different versions of MCM, namely v3.1 vs. v3.3.1, but the main differences concerned updates to mechanisms for some biogenic VOCs (Jenkin et al., 2015), whose reactivities were not used in the comparison. The five outliers shown in Figure 6 (*n*-nonane, *n*-decane, *n*-octane, benzene, and styrene) are compounds with both strong $O_3$ formation and $O_3$ inhibition characteristics and relatively low net reactivities that are highly variable because the relative importance of these opposing effects depends on environmental conditions. The IRs for the

former three long-chain alkanes reflect a balance between the positive effects on $O_3$ due to the conversions of NO to $NO_2$ by the radicals they form, and also due to the reactivity of its products, and the negative effects caused by radical inhibition (organic nitrates). This is also the case for benzene and styrene, whose major degradation products are phenol and benzaldehyde, respectively, which are strong inhibitors of $O_3$ formation under specific conditions.

    The comparison illustrated that SAPRC and MCM diverged somewhat with respect to the understanding of atmospheric

oxidation of a few compounds, although they shared a common representation of the reaction mechanisms of many other VOCs (Figures 5 and S4b) (Carter, 2010; Jenkin et al., 2003; Saunders et al., 2003). Furthermore, the IR scales showed a

strong dependence on the adopted chemical mechanisms, which indicates the need for accurate representation of explicit chemical mechanisms.

**3.3 Other possible reasons for the IR discrepancy between China and the U.S. scenarios**

We designed a series of sensitivity tests to deduce other possible reasons for the IR discrepancy between China and the U.S., and also examined the effects of environmental conditions on the IR scales. As shown in Table 2, the effects of HONO levels on the IR scales were insignificant (within 7 %), whereas the VOC/NOx ratio and VOC composition were important factors. One quarter of the base VOC/NOx ratios (both initial and emitted VOCs multiplied by 0.25, while other factors were kept unchanged) in the Guangzhou scenarios tended to increase the overall IR values (by 19 % and 26 % under the MIR and

MOR scenarios, respectively). This factor was excluded as the possible reason for the discrepancy between Guangzhou and the U.S. owing to the lower VOC/NOx ratios in Guangzhou (0.99 vs. 6.60 in the U.S. average scenarios). The effects of the VOC composition change on the IR scales were double-edged. With the same total VOC mass, higher proportions of the reactive VOC class (such as alkenes) would increase the relative availability of VOCs and reduce the importance of VOCs to $O_3$ formation. For example, doubling the base proportion of the alkene class would decrease the overall IR values (by -10 %

and -12 % under the MIR and MOR scenarios, respectively), but 1.5 times the base proportion of the alkane class would increase the overall IR values (by 20 % and 25 % under the MIR and MOR scenarios, respectively). One quarter of the base proportion of the OVOC class would slightly increase the overall IR values (by 8 % and 2 % under the MIR and MOR scenarios, respectively). Owing to the lack of detailed information on U.S. scenarios, we could not directly quantify the effect of a single variable, but inferred that the higher alkene and OVOC proportions and lower alkane proportions in the

Guangzhou scenarios (Figure 1) were possible reasons for the lower localized IR values than those of the U.S. It is also possible that the dilution process served as an $O_3$ loss pathway and reduced the VOC reactivity. Doubling the base proportions of the aromatic class would increase the overall IR values under the MIR scenarios (by 3 %) but decrease the overall IR values under the MOR scenarios (by -8 %). This was reasonable considering that aromatics and their products are major NOx sinks under low NOx availability conditions. The higher aromatic proportions in emission-based inputs (24 % vs.

20 % in the observation-based inputs; Figure 1) partly explained why the MIR values obtained from emission-based inputs were higher than those obtained from observation-based inputs, but the MOR and EBIR values were lower. The abovementioned chemical environmental conditions would affect the IR magnitudes to different degrees.

We also examined the effects of other environmental conditions on the IR scales (Table 2). The weakening of photolysis rates by 50 % could significantly reduce the overall MIR (by -31 %; $p < 0.01$) and MOR values (by -13 %; $p < 0.01$),

particularly the IR of aldehydes (such as propionaldehyde and glyoxal), the photolysis of which behaved as a dominant primary radical source and was more susceptible to photolysis intensity change. Another environmental condition considered was the effusion of VOCs from the urban scale to the regional scale. A regional scenario (the model integration time lasted for 3 days, whereas the other factors were kept unchanged; indicated as "3 days" in Table 2) was designed to evaluate the

relative importance of VOCs and NOx as well as of different VOC species to $O_3$ formation in regional scales. The overall

MIR and MOR values for the 116 VOCs changed by -5 % ($p = 0.07$) and -61 % ($p < 0.01$), respectively, implying that the NOx emissions rather than VOC emissions from upwind urban sources would play more important roles in $O_3$ concentrations over large regional scales. The NOx conditions under MIR scenarios are higher than under MOR scenarios, therefore, the MORs dropped much faster than MIRs in 3-day regional scenarios. Along the 3-day scale, the reactive VOC groups (such as alkenes) with a short lifetime were rapidly consumed in urban scales, but their IRs showed fast downward

trends in 3-day regional scales (e.g., the median rank of alkenes dropped by -11 and -27 under the MIR and MOR scenarios, respectively). In contrast, the role of unreactive VOC groups became more important, which would build up and undergo extensive photochemical reactions along the 3-day scale (Stockwell et al., 2001). Besides, the unreactive VOC groups (such as alkanes) and their oxidation products would indirectly impact the IRs of other compounds by exerting effects on radical recycling. Taken together, the IRs for unreactive VOC groups showed slowly downward or even upward trends in 3-day

scenarios than 10-hour scenarios. The opposite trends explained why $R^2$ were relatively small for some VOC groups between 3-day and 10-hour scenarios. These two conditions were not considered as possible reasons for the IR discrepancy between Guangzhou and the U.S. because both studies assumed a clear sky and set similar integration times (i.e., 10 hours).

We also conducted similar analyses on RR scales, and the results confirmed that environmental conditions exert large impact on the RR magnitudes (Table S5). Considering that the IR scales are more sensitive to environmental conditions with the

decrease in NOx availability, it is necessary to localize the $O_3$ reactivity scales for VOCs using realistic scenarios under rapidly changing environments (especially under low NOx conditions).

### 3.4 Applications for Chinese cities

MIR-Carter 2010 is the most extensively used scale to quantify the OFP and identify the key reactive VOCs in Chinese cities. To better address the aforementioned concerns regarding the IR application, two sets of model inputs were designed using

the megacity of Guangzhou to localize the IR scales under three specified NOx conditions. We revealed the strong dependence of IR scales on the adopted chemical mechanisms and demonstrated that both emission-based and observation-based inputs are suitable for the MIR calculation. Because $O_3$ formation in most cities in China (and many other countries) is limited by VOCs, it is essential to clearly elucidate how to apply IR and OFP scales to deal with VOC control. Here, we provide recommendations on the application from three perspectives.

**On IR localization:** With a given chemical mechanism, the chemical environmental conditions would affect the MIR magnitudes but exert little effect on the MIR ranks of most VOCs, thereby demonstrating that a general MIR scale can be roughly used to identify the reactive VOCs in most VOC-limited areas. However, as the NOx availability decreased, the IR scale became more sensitive to the environmental conditions. It would be better to localize the MOR and EBIR scales over mixed-limited or NOx-limited areas. However, with a prescribed NOx availability, the IR scales were more sensitive to the

chemical mechanisms than to chemical environmental conditions, thereby implying that the establishment of an explicit and accurate chemical mechanism is more important than IR localization.

**On the role of emission data and observational data for IR calculation:** The discrepancy between the IR scales obtained from the emission-based and observation-based inputs also obeyed the abovementioned "NOx availability" rule. Under high NOx conditions, the fairly good consistency between the MIR scales obtained from the emission-based and observation-

based inputs suggested that both inputs can accurately estimate the actual environment and are suitable for the MIR calculation. However, the discrepancy between the MOR and EBIR magnitudes obtained from the two inputs was relatively large and cannot be overlooked, thereby underlining the need to use realistic scenarios for IR calculations, especially under low NOx conditions.

**On the OFP quantification:** The observed VOC data were suitable for calculating the MIR scale, but caution should be

taken when using them to quantify the OFP by multiplying the IR by observed concentrations. For example, it is unreasonable to use the observed VOC data during periods with intense photochemistry to quantify the OFP, which would underestimate the importance of reactive VOCs to $O_3$ formation. For this purpose, we recommend using observed VOC data in the morning for OFP quantification. Furthermore, some VOCs (such as formaldehyde and acetaldehyde) have both primary and secondary sources. The secondary contributions should be removed for the OFP quantification, otherwise the

contributions of secondary sources will be counted twice. Besides, in comparison with the absolute OFP, the ROFP is a more reliable scale (Bowman and Seinfeld, 1995; Chang and Rudy, 1990; Japar et al., 1991; Russell et al., 1995). The ROFP is calculated as follows:

$$\mathrm{ROFP}_i = \frac{[\mathrm{VOC}_i] * \mathrm{IR}_i}{[\mathrm{VOC}_r] * \mathrm{IR}_r} \quad (2)$$

where $\mathrm{ROFP}_i$ represents the ROFP of $\mathrm{VOC}_i$; $\mathrm{IR}_i$ and $\mathrm{IR}_r$ represent the IRs of $\mathrm{VOC}_i$ and the reference VOC, respectively; and

$[\mathrm{VOC}]_i$ and $[\mathrm{VOC}]_r$ represent the observed concentrations (or emission quantity/emission rate) of $\mathrm{VOC}_i$ and the reference VOC, respectively. Ethene is recommended as a reference species considering its well-defined chemical reaction pathways and importance to $O_3$ formation, and its use as the reference species in other studies (e.g., Derwent et al., 2010). The advantage of the ROFP is that it can minimize the effect of background variability.

In the future, more efforts are required to develop more comprehensive and realistic $O_3$ reactivity scales for VOCs. The

following directions are recommended: 1) model scenarios with realistic representations of physical factors, such as photolysis intensity, are needed to reduce the uncertainty caused by simple parameterizations; 2) emission inventories with higher temporal and spatial resolutions and detailed VOC emission profiles are needed to reduce the uncertainty introduced by primary emissions; and 3) accurate representation of the chemical mechanisms within models is crucial. The chemical mechanism should represent the state of the science and appropriately represent all of the important VOCs present. In

addition, because the mechanism significantly affects the IR results, it is important that the same chemical mechanism be used when assessing the effects of environmental conditions and localization on IR scales or OFP estimates in China or

elsewhere. To make comprehensive comparisons, a major work focusing on VOC reactivity scales obtained from the U.S. conditions vs. Chinese conditions using the same and different chemical mechanisms is needed.

## 4 Conclusions

We designed two sets of model inputs (emission-based and observation-based) to localize the IR scales for VOCs in Guangzhou, southern China, and compared the localized IR results with those of extensively used U.S. IR scales. With a prescribed NOx availability, the IR scales were more sensitive to chemical mechanisms (MCM vs. SAPRC) than to chemical environmental conditions. With a given chemical mechanism, the discrepancy in IR ranks between China and the U.S. was not large, thereby implying that the establishment of an accurate chemical mechanism is more important than IR localization.

However, although there was a fairly good consistency between the localized MIR scales obtained from the emission-based and observation-based inputs, the IR discrepancy increased as the NOx availability decreased, thereby implying the necessity to localize IR scales over mixed-limited or NOx-limited areas and the need for realistic scenarios in rapidly changing environments. The localized IR results in Guangzhou were taken as a case to clearly elucidate how to apply IR and OFP scales to deal with reactive VOC identification, which has great implications for VOC control in China and other countries

suffering from serious $O_3$ pollution. Considering the huge impact of chemical mechanisms and environmental conditions on the IR scales, more systematic comparisons focusing on VOC reactivity scales obtained from the U.S. vs. Chinese conditions using the same and different chemical mechanisms are needed.

## Data availability

The emission- and observation-based inputs and model output used in the present study can be accessed from

http://dx.doi.org/10.17632/2y752t39yn.1 (Zhang and Xue, 2021). The code for the MCM chemical box model can be downloaded from the F0AM website (https://github.com/AirChem/F0AM).

## Author contributions

LK designed the research. CP and YW conducted the field campaigns. YZ conducted the chemical box modeling analyses and wrote the paper. TC and JM helped in the creation of model. LK, WPLC, QZ, and WW helped in the interpretation of

results and revised the original manuscript.

## Competing interests

The authors declare that they have no conflict of interest.

## Acknowledgements

We thank R. G. Derwent for providing the MIR/Ethene data of California scenarios, NASA for providing the platform of F0AM, and the University of Leeds for providing the Master Chemical Mechanism (version 3.3.1). We also thank the Tsinghua University for providing the MEIC emission inventory, and the Peking University for providing the biogenic emission inventory. This work was funded by the National Natural Science Foundation of China (41922051), Shandong Provincial Science Foundation for Distinguished Young Scholars (ZR2019JQ09), the Jiangsu Collaborative Innovation Center for Climate Change, the Taishan Scholars (ts201712003), and Science and Technology Program of Guangdong Province (Science and Technology Innovation Platform Category; 2019B121201002).

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

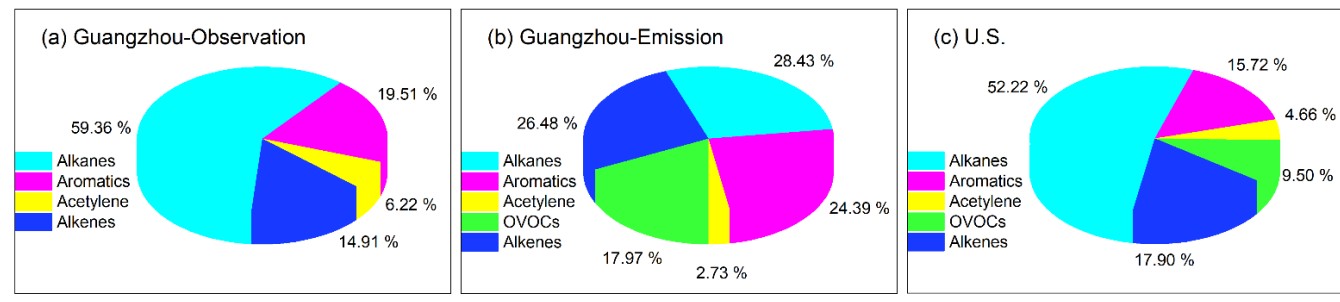

**Figure 1:** VOC compositions (ppbv ppbv$^{-1}$) in (a) observation-based inputs of Guangzhou scenarios, (b) emission-based inputs of Guangzhou scenarios, and (c) U.S. scenarios. The U.S. scenario data are taken from Carter (1994b).

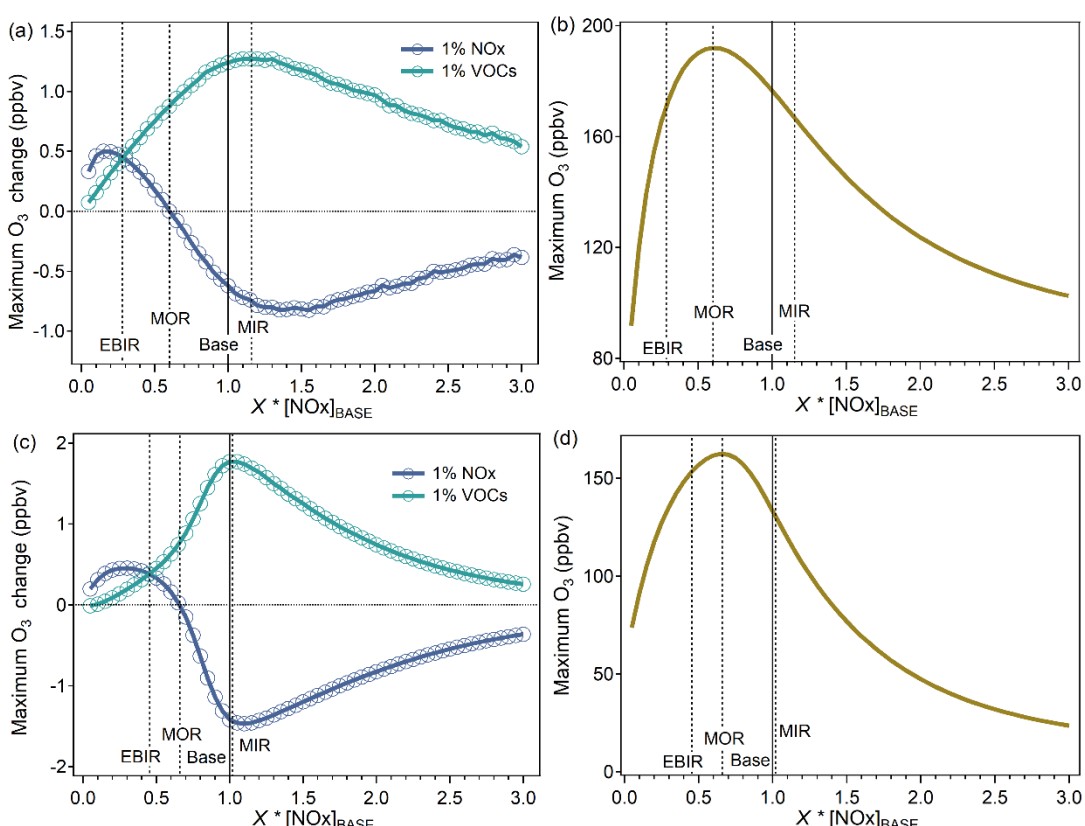

**Figure 2:** Identification of the MIR, MOR, and EBIR scenarios. The model-simulated ((a) and (c)) maximum hourly $O_3$ change caused by
1 % increases in VOC or NOx inputs and ((b) and (d)) maximum hourly $O_3$ as a function of $X \times$ base case NOx inputs (represented as
$[NOx]_{BASE}$). The upper panel shows the observation-based inputs and the lower panel shows the emission-based inputs. The MIR scenarios
represent the NOx conditions in which VOC control has the greatest effect on $O_3$ formation. The MOR scenarios represent the NOx
conditions that are optimal for $O_3$ formation. The EBIR scenarios represent NOx conditions in which $O_3$ formation transits from a mixed-
limited regime to an NOx-limited regime.

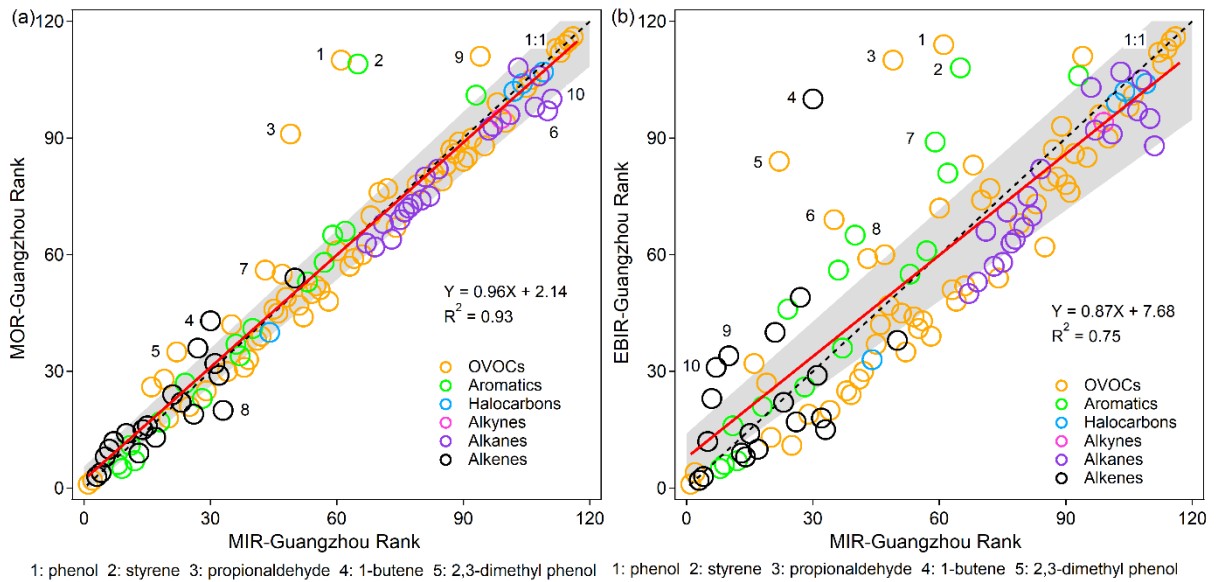


**Figure 3:** Correlations between the emission-based IR ranks for 116 VOC species under MIR scenarios versus (a) MOR scenarios and (b) EBIR scenarios. The top 10 VOC species with a relatively large rank change are marked with numbers (shown below the individual panels). The IR scales for VOCs are ranked in descending order (from the highest IR value (rank = 1) to the lowest IR value (rank = 116)). The gray shaded areas indicate the least square linear regression line with ± 95 % confidence and the black dashed line represents the 1:1 line.


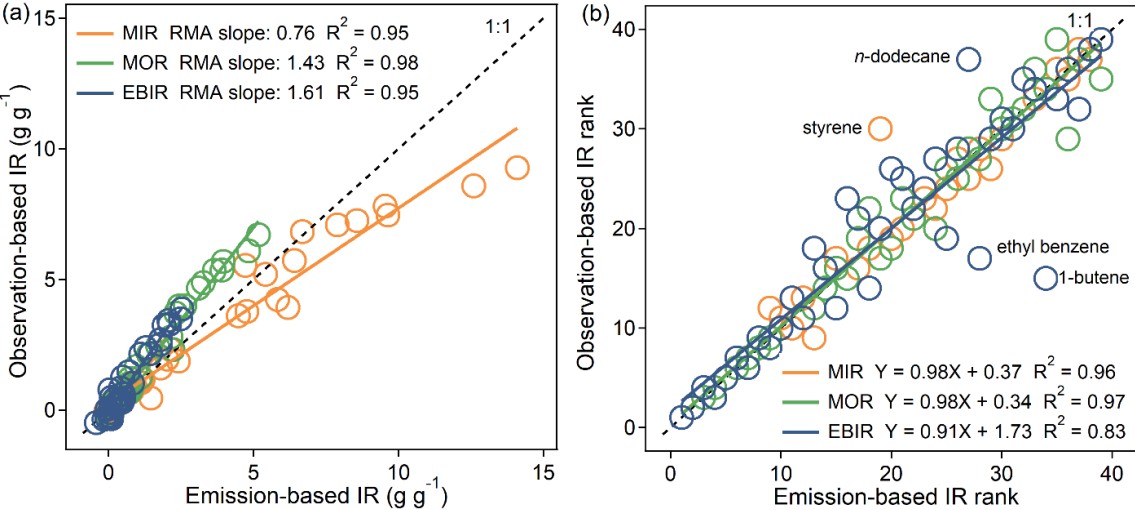

**Figure 4:** Comparison of IR (a) values and (b) ranks for 39 common VOC species between both emission-based inputs and observation-based inputs. The RMA represents reduced major axis. The IR scales for VOCs are ranked in descending order (from the highest IR value (rank = 1) to the lowest IR value (rank = 39)). The black dashed line represents the 1:1 line.

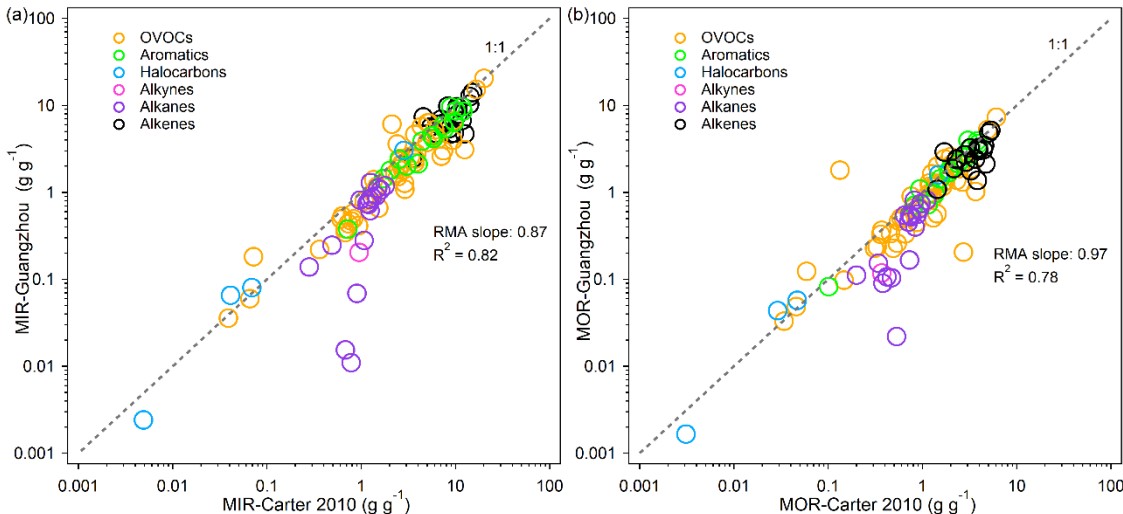

**Figure 5:** Comparison of emission-based (a) MIR and (b) MOR scales for 111 common VOC species between Guangzhou and the U.S. The panels are in log scale, and only positively reactive VOCs are shown. The grey dashed line represents the 1:1 line. The MIR-Carter 2010 and MOR-Carter 2010 data are taken from Carter et al. (2010).

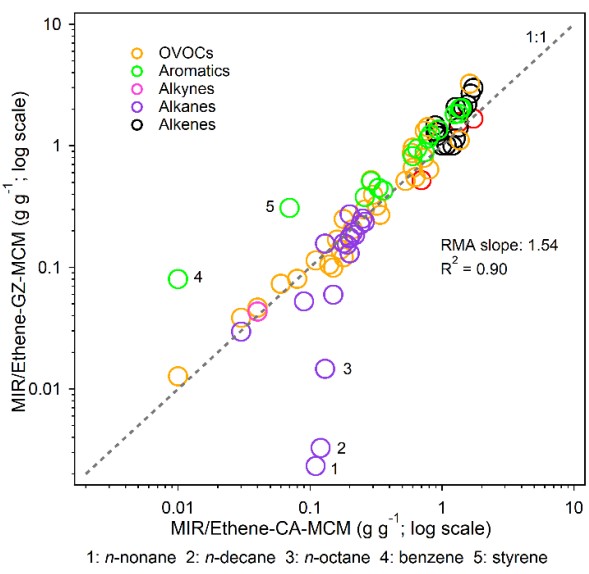

1: *n*-nonane  2: *n*-decane  3: *n*-octane  4: benzene  5: styrene

**Figure 6:** Comparison of the MIR/Ethene (the MIR value of a given VOC divided by the MIR value of ethene) values for 79 common VOC species between Guangzhou and California. Only positively reactive VOCs are shown, and the top five VOC species with a relatively large reactivity value change (shown below the panel) are marked with numbers. The grey dashed line represents the 1:1 line. The MIR-Ethene-CA-MCM data are taken from Derwent et al. (2010).

**Table 1.** Tabulation of IR (g O$_3$/g VOC) for VOCs under three specified NOx conditions and base NOx conditions. "EMI" represents emission–based inputs, while "OBS" represents observation–based inputs.

| Species | MIR | MOR | EBIR | Base IR |
|---------|-----|-----|------|---------|

|  | EMI | OBS | EMI | OBS | EMI | OBS | EMI | OBS |
|---|---|---|---|---|---|---|---|---|
| ethane | 0.14 | 0.19 | 0.11 | 0.16 | 0.08 | 0.09 | 0.14 | 0.19 |
| propane | 0.25 | 0.25 | 0.15 | 0.17 | 0.08 | 0.07 | 0.25 | 0.25 |
| *n*-butane | 0.75 | 0.83 | 0.54 | 0.63 | 0.35 | 0.33 | 0.76 | 0.83 |
| *i*-butane | 0.83 | 0.91 | 0.55 | 0.71 | 0.37 | 0.40 | 0.85 | 0.92 |
| *n*-pentane | 0.87 | 0.94 | 0.56 | 0.66 | 0.30 | 0.29 | 0.89 | 0.93 |
| *i*-pentane | 0.91 | 0.93 | 0.64 | 0.71 | 0.44 | 0.38 | 0.93 | 0.93 |
| *n*-hexane | 0.62 | 0.75 | 0.40 | 0.48 | 0.20 | 0.17 | 0.62 | 0.73 |
| 2,2-dimethyl butane | 0.74 | 0.67 | 0.46 | 0.48 | 0.27 | 0.23 | 0.75 | 0.67 |
| 2,3-dimethyl butane | 0.81 | 0.75 | 0.54 | 0.51 | 0.37 | 0.26 | 0.82 | 0.73 |
| 2-methyl pentane | 1.12 | 1.04 | 0.65 | 0.67 | 0.35 | 0.29 | 1.13 | 0.99 |
| 3-methyl pentane | 1.19 | 1.18 | 0.81 | 0.88 | 0.52 | 0.46 | 1.21 | 1.18 |
| cyclohexane | 1.29 | - | 0.81 | - | 0.66 | - | 1.27 | - |
| *n*-heptane | 0.28 | 0.34 | 0.17 | 0.11 | 0.03 | -0.08 | 0.28 | 0.30 |
| *n*-octane | 0.07 | -0.0001 | -0.03 | -0.25 | -0.16 | -0.35 | 0.06 | -0.07 |
| *n*-nonane | 0.01 | -0.02 | 0.02 | -0.23 | -0.06 | -0.30 | 0.0003 | -0.09 |
| *n*-decane | 0.02 | 0.06 | 0.10 | -0.14 | 0.04 | -0.22 | 0.01 | -0.003 |
| *n*-undecane | -0.03 | -0.003 | 0.11 | -0.20 | 0.07 | -0.26 | -0.04 | -0.07 |
| *n*-dodecane | -0.10 | -0.04 | 0.09 | -0.27 | 0.10 | -0.33 | -0.12 | -0.12 |
| 2-methyl hexane | 0.73 | - | 0.52 | - | 0.32 | - | 0.74 | - |
| 3-methyl hexane | 1.06 | 1.04 | 0.75 | 0.69 | 0.47 | 0.31 | 1.07 | 1.00 |
| ethene | 4.71 | 5.54 | 2.43 | 3.98 | 1.44 | 2.16 | 4.75 | 5.42 |
| propene | 6.68 | 6.83 | 3.09 | 4.67 | 1.78 | 2.69 | 6.66 | 6.66 |
| 1-butene | 4.77 | 3.77 | 1.37 | 2.14 | 0.03 | 0.77 | 4.68 | 3.49 |
| *i*-butene | 4.75 | - | 1.88 | - | 1.03 | - | 4.69 | - |
| *cis*-2-butene | 12.64 | 8.58 | 4.83 | 6.09 | 2.48 | 3.49 | 12.52 | 8.41 |
| *trans*-2-butene | 14.11 | 9.27 | 5.17 | 6.71 | 2.52 | 3.87 | 13.97 | 9.42 |
| 1,3-butadiene | 4.73 | - | 2.12 | - | 1.36 | - | 4.66 | - |
| 1-pentene | 6.18 | 3.93 | 2.22 | 2.31 | 0.82 | 1.02 | 6.09 | 3.68 |
| 3-methyl-1-butene | 6.93 | - | 2.71 | - | 1.46 | - | 6.85 | - |
| 2-methyl-1-butene | 6.08 | - | 2.38 | - | 1.24 | - | 6.02 | - |
| 2-methyl-2-butene | 10.31 | - | 3.46 | - | 1.53 | - | 10.16 | - |
| *cis*-2-pentene | 9.69 | - | 3.10 | - | 0.97 | - | 9.54 | - |
| *trans*-2-pentene | 9.45 | - | 3.01 | - | 0.93 | - | 9.31 | - |
| isoprene | 7.89 | 7.09 | 3.26 | 4.88 | 1.78 | 2.51 | 7.82 | 6.87 |
| 1-hexene | 5.69 | - | 2.44 | - | 1.37 | - | 5.61 | - |
| *cis*-2-hexene | 9.83 | - | 3.25 | - | 1.20 | - | 9.68 | - |
| *trans*-2-hexene | 5.44 | - | 1.77 | - | 0.66 | - | 5.36 | - |
| *β*-pinene | 2.46 | - | 1.08 | - | 0.84 | - | 2.37 | - |
| limonene | 7.36 | - | 2.91 | - | 1.87 | - | 7.18 | - |

| | | | | | | | | |
|---|---|---|---|---|---|---|---|---|
| benzene | 0.38 | 0.49 | 0.08 | 0.27 | -0.06 | 0.02 | 0.38 | 0.46 |
| toluene | 2.12 | 2.33 | 0.96 | 1.67 | 0.41 | 0.84 | 2.14 | 2.3 |
| ethyl benzene | 2.02 | 1.95 | 0.72 | 1.21 | 0.09 | 0.48 | 2.02 | 1.87 |
| *m*-xylene | 6.41 | 5.72 | 2.63 | 4.00 | 1.27 | 2.38 | 6.38 | 5.47 |
| *o*-xylene | 5.41 | 5.20 | 2.31 | 3.72 | 1.11 | 2.14 | 5.41 | 5.06 |
| *p*-xylene | 4.17 | - | 1.82 | - | 0.87 | - | 4.17 | - |
| styrene | 1.45 | 0.46 | -0.04 | -0.17 | -0.43 | -0.48 | 1.41 | 0.30 |
| *n*-propyl benzene | 1.79 | 1.61 | 0.71 | 1.01 | 0.21 | 0.42 | 1.79 | 1.56 |
| *i*-propyl benzene | 2.41 | 1.87 | 1.09 | 1.26 | 0.5 | 0.62 | 2.43 | 1.82 |
| *m*-ethyl toluene | 5.83 | 4.23 | 2.15 | 2.80 | 0.70 | 1.47 | 5.8 | 3.98 |
| *o*-ethyl toluene | 4.47 | 3.61 | 1.69 | 2.46 | 0.48 | 1.26 | 4.46 | 3.49 |
| *p*-ethyl toluene | 3.87 | - | 1.43 | - | 0.36 | - | 3.87 | - |
| 1,2,3-trimethyl benzene | 8.57 | 7.25 | 3.65 | 5.35 | 1.91 | 3.24 | 8.54 | 7.20 |
| 1,2,4-trimethyl benzene | 9.54 | 7.81 | 3.95 | 5.69 | 2.07 | 3.43 | 9.48 | 7.69 |
| 1,3,5-trimethyl benzene | 9.64 | 7.47 | 3.91 | 5.36 | 2.10 | 3.29 | 9.54 | 7.26 |
| 1,3-dimethyl-5-ethyl benzene | 8.64 | - | 3.22 | - | 1.42 | - | 8.55 | - |
| formaldehyde | 6.71 | - | 2.21 | - | 0.97 | - | 6.61 | - |
| methanol | 0.38 | - | 0.23 | - | 0.14 | - | 0.39 | - |
| formic acid | 0.06 | - | 0.05 | - | 0.04 | - | 0.06 | - |
| ethylene oxide | 0.04 | - | 0.03 | - | 0.03 | - | 0.04 | - |
| acetaldehyde | 4.10 | - | 1.88 | - | 1.15 | - | 4.06 | - |
| ethanol | 0.66 | - | 0.45 | - | 0.29 | - | 0.67 | - |
| dimethyl ether | 0.57 | - | 0.47 | - | 0.38 | - | 0.58 | - |
| glyoxal | 3.11 | - | 1.02 | - | 0.42 | - | 3.06 | - |
| acetic acid | 0.35 | - | 0.24 | - | 0.17 | - | 0.35 | - |
| acrolein | 3.01 | - | 1.31 | - | 0.84 | - | 2.97 | - |
| propionaldehyde | 2.60 | - | 0.21 | - | -0.55 | - | 2.49 | - |
| acetone | 0.22 | - | 0.10 | - | 0.05 | - | 0.22 | - |
| *i*-propyl alcohol | 0.49 | - | 0.33 | - | 0.23 | - | 0.50 | - |
| *n*-propyl alcohol | 1.86 | - | 0.85 | - | 0.29 | - | 1.87 | - |
| methyl glyoxal | 15.27 | - | 5.25 | - | 2.31 | - | 15.1 | - |
| methyl acetate | 0.18 | - | 0.12 | - | 0.08 | - | 0.19 | - |
| propylene glycol | 2.45 | - | 1.28 | - | 0.74 | - | 2.48 | - |
| dimethoxy methane | 0.41 | - | 0.33 | - | 0.25 | - | 0.42 | - |
| crotonaldehyde | 3.97 | - | 1.85 | - | 1.19 | - | 3.92 | - |
| methacrolein | 5.23 | - | 2.21 | - | 1.28 | - | 5.17 | - |
| 2-methyl propanal | 6.29 | - | 2.14 | - | 1.11 | - | 6.16 | - |
| butanal | 4.54 | - | 1.38 | - | 0.33 | - | 4.43 | - |
| methyl ethyl ketone | 1.18 | - | 0.51 | - | 0.27 | - | 1.17 | - |
| *i*-butyl alcohol | 2.28 | - | 1.18 | - | 0.74 | - | 2.29 | - |

| | | | | | | | |
|---|---|---|---|---|---|---|---|
| n-butyl alcohol | 1.27 | - | 0.57 | - | 0.19 | - | 1.27 | - |
| sec-butyl alcohol | 1.40 | | 0.90 | - | 0.60 | - | 1.43 | - |
| diethyl ether | 2.42 | - | 1.36 | - | 0.92 | - | 2.43 | - |
| biacetyl | 20.42 | - | 7.30 | - | 3.42 | - | 20.25 | - |
| ethyl acetate | 0.54 | - | 0.37 | - | 0.24 | - | 0.55 | - |
| 1-methoxy-2-propanol | 1.51 | - | 0.95 | - | 0.66 | - | 1.53 | - |
| 2-ethoxy-ethanol | 2.20 | - | 1.16 | - | 0.79 | - | 2.19 | - |
| diethylene glycol | 2.6 | - | 1.22 | - | 0.69 | - | 2.61 | - |
| 3-methylbutanal | 3.75 | - | 1.67 | - | 1.09 | - | 3.70 | - |
| pentanal | 6.27 | - | 2.57 | - | 1.49 | - | 6.15 | - |
| methyl tert-butyl ether | 0.44 | - | 0.33 | - | 0.24 | - | 0.45 | - |
| i-propyl acetate | 0.80 | - | 0.51 | - | 0.34 | - | 0.81 | - |
| propyl acetate | 0.47 | - | 0.23 | - | 0.07 | - | 0.47 | - |
| phenol | 1.80 | - | -0.26 | - | -1.37 | - | 1.82 | - |
| 2-hexanone | 2.71 | - | 1.05 | - | 0.41 | - | 2.67 | - |
| cyclohexanone | 0.97 | - | 0.66 | - | 0.52 | - | 0.97 | - |
| hexanal | 5.72 | - | 2.42 | - | 1.53 | - | 5.61 | - |
| 4-methyl-2-pentanone | 2.78 | - | 1.32 | - | 0.78 | - | 2.77 | - |
| 4-methyl-2-pentanol | 2.03 | - | 1.27 | - | 0.83 | - | 2.07 | - |
| hexanol | 2.13 | - | 1.17 | - | 0.77 | - | 2.12 | - |
| n-butyl acetate | 0.50 | - | 0.26 | - | 0.1 | - | 0.50 | - |
| 2-butoxy-ethanol | 1.08 | - | 0.51 | - | 0.25 | - | 1.07 | - |
| benzaldehyde | -1.02 | - | -1.04 | - | -1.17 | - | -1.04 | - |
| acetophenone | 0.35 | - | -0.27 | - | -0.56 | - | 0.33 | - |
| benzyl alcohol | -0.33 | - | -0.74 | - | -0.97 | - | -0.36 | - |
| 2-Methylbenzaldehyde | -1.19 | - | -1.36 | - | -1.65 | - | -1.22 | - |
| 3-Methylbenzaldehyde | -1.16 | - | -1.24 | - | -1.50 | - | -1.18 | - |
| 4-methylbenzaldehyde | -0.65 | - | -0.41 | - | -0.51 | - | -0.66 | - |
| heptanal | 4.63 | - | 2.01 | - | 1.28 | - | 4.54 | - |
| 5-methyl-2-hexanone | 3.60 | - | 1.64 | - | 1.03 | - | 3.56 | - |
| 2,3-dimethyl phenol | 6.08 | - | 1.80 | - | 0.19 | - | 6.05 | - |
| 3-octanol | 1.65 | - | 0.97 | - | 0.62 | - | 1.66 | - |
| dichloromethane | 0.07 | - | 0.04 | - | 0.03 | - | 0.07 | - |
| acetylene | 0.20 | 0.22 | 0.12 | 0.17 | 0.07 | 0.09 | 0.21 | 0.22 |
| vinyl chloride | 3.03 | - | 1.60 | - | 0.94 | - | 3.06 | - |
| 1,1-dichloroethane | 0.08 | - | 0.06 | - | 0.04 | - | 0.08 | - |
| 1,1,1-trichloroethane | 0.002 | - | 0.002 | - | 0.001 | - | 0.002 | - |

**Table 2a.** Dependence of emission–based MIR scales for major VOC groups on major environmental conditions. The base MIR scales were used as a benchmark (x-axis). Eight sensitivity scenarios: (2) 0.25*VOCs: both initial and emitted VOCs multiplied by 0.25, while other factors were kept unchanged, (3)-(6) X*VOC group: both initial and emitted VOC species contained in the target VOC group multiplied by X, and other VOC compositions increased or decreased to keep VOCs/NOx ratios constant, while other factors were kept unchanged, (7) HONO: both initial and emitted HONO were set to zero, and only the gas phase formation pathway (OH + NO = HONO) of HONO was included in the model, (8) 0.5*J: the photolysis rate was reduced to a half, while other factors were kept unchanged, (9) 3 days: the model integration time lasted for 3 days, while other factors were kept unchanged. Refer to Table S3 for the detailed speciation of VOCs.

| VOC group | 0.25*VOCs | | 0.25*OVOCs | | 1.5*Alkanes | | 2*Aromatics | | 2*Alkenes | | HONO | | 0.5*J | | 3 days | |
|---|---|---|---|---|---|---|---|---|---|---|---|---|---|---|---|---|
| | RMA slope | $R^2$ | RMA slope | $R^2$ | RMA slope | $R^2$ | RMA slope | $R^2$ | RMA slope | $R^2$ | RMA slope | $R^2$ | RMA slope | $R^2$ | RMA slope | $R^2$ |
| Aldehydes | 1.17 | 0.99 | 1.11 | 1.00 | 1.27 | 1.00 | 1.05 | 1.00 | 0.87 | 1.00 | 1.10 | 1.00 | 0.77 | 0.98 | 0.97 | 0.83 |
| Alkanes | 1.36 | 0.99 | 1.00 | 1.00 | 1.11 | 0.99 | 0.97 | 1.00 | 0.96 | 1.00 | 1.00 | 1.00 | 0.56 | 0.92 | 1.47 | 0.43 |
| Alkenes | 1.01 | 0.97 | 1.12 | 1.00 | 1.22 | 1.00 | 1.17 | 0.99 | 0.84 | 0.99 | 1.14 | 1.00 | 0.78 | 0.97 | 0.61 | 0.64 |
| Aromatics | 1.33 | 1.00 | 1.02 | 1.00 | 1.12 | 1.00 | 0.91 | 1.00 | 0.95 | 1.00 | 1.00 | 1.00 | 0.57 | 0.98 | 0.97 | 0.95 |
| Ketones | 1.38 | 1.00 | 1.00 | 1.00 | 1.14 | 1.00 | 0.92 | 1.00 | 0.96 | 1.00 | 1.00 | 1.00 | 0.51 | 0.98 | 1.77 | 0.90 |
| TVOC | 1.19 | 0.99 | 1.08 | 1.00 | 1.20 | 1.00 | 1.03 | 0.99 | 0.90 | 1.00 | 1.07 | 1.00 | 0.69 | 0.97 | 0.95 | 0.81 |

**Table 2b.** The same as Table 2a but for MOR scales.

| VOC group | 0.25*VOCs | | 0.25*OVOCs | | 1.5*Alkanes | | 2*Aromatics | | 2*Alkenes | | HONO | | 0.5*J | | 3 days | |
|---|---|---|---|---|---|---|---|---|---|---|---|---|---|---|---|---|
| | RMA slope | $R^2$ | RMA slope | $R^2$ | RMA slope | $R^2$ | RMA slope | $R^2$ | RMA slope | $R^2$ | RMA slope | $R^2$ | RMA slope | $R^2$ | RMA slope | $R^2$ |
| Aldehydes | 1.27 | 0.99 | 1.05 | 1.00 | 1.33 | 0.99 | 0.90 | 1.00 | 0.86 | 1.00 | 1.10 | 1.00 | 0.98 | 0.95 | 0.29 | 0.01 |
| Alkanes | 1.35 | 0.98 | 0.98 | 1.00 | 1.10 | 0.97 | 0.95 | 0.98 | 0.95 | 0.99 | 0.98 | 1.00 | 0.69 | 0.81 | 0.90 | 0.31 |
| Alkenes | 1.11 | 0.98 | 1.04 | 1.00 | 1.27 | 0.98 | 1.00 | 0.99 | 0.84 | 0.98 | 1.13 | 0.99 | 1.00 | 0.91 | 0.58 | 0.06 |
| Aromatics | 1.36 | 1.00 | 0.99 | 1.00 | 1.15 | 1.00 | 0.87 | 1.00 | 0.93 | 1.00 | 1.00 | 1.00 | 0.69 | 0.98 | 0.51 | 0.80 |
| Ketones | 1.41 | 0.99 | 0.97 | 1.00 | 1.13 | 1.00 | 0.93 | 1.00 | 0.94 | 1.00 | 1.00 | 1.00 | 0.60 | 0.98 | 0.71 | 0.09 |
| TVOC | 1.26 | 0.99 | 1.02 | 1.00 | 1.25 | 0.99 | 0.92 | 1.00 | 0.88 | 1.00 | 1.07 | 1.00 | 0.87 | 0.97 | 0.39 | 0.42 |

