# Peer review of "Development of Ozone Reactivity Scales for Volatile Organic Compounds in a Chinese Megacity"

_Atmospheric Chemistry and Physics, 2021_

## Author Response (AR1)

**Response to Anonymous Referee #1**

*General Comments:*

*Zhang et al. describe a comprehensive analysis of reactivity scales for a Chinese megacity, Guangzhou. They use two methods to characterize base conditions for their Master Chemical Mechanism (MCM) box model. In one, model inputs are based on observations and in the other they are based on emissions. Four reactivity scales were determined using base conditions, maximum incremental reactivity (MIR), maximum ozone reactivity (MOR) and equal benefit incremental reactivity (EBIR) conditions. MOR and MIR reactivity scales estimated for Guangzhou were compared with the corresponding scales for USA conditions using the same chemical mechanism and with the SAPRC-07 mechanism. Sensitivity tests were performed to investigate the influence of environmental conditions on the estimated reactivity scales.*

*This is a comprehensive study which is well worth publishing in ACP. It should provide the essential data required in the formulation of strategies for tackling elevated ozone levels across China. I particularly liked Figure 2 and how the study links the MIR scale to VOC-limited conditions, MOR to mixed and EBIR to NOx-limited conditions.*

**Response:** we thank the reviewer for the positive comments and constructive suggestions, which are much helpful for improving the original manuscript. We have carefully considered all of these comments and revised the manuscript accordingly. Below we provide the original reviewer's comments in black italic, with our response and changes in the manuscript in blue and red, respectively.

*The study fulfils an important service in providing in Table 1 the IRs for all 116 VOC species in the MCM under the four sets of conditions. Elsewhere in the study more use should be made of relative incremental reactivities, by expressing them relative to a specific VOC such as ethene. Zhang et al. introduce the concept as MIR/Ethene though much more use could be made than the brief mention in section 3.2 (lines 227 onwards). The advantage of relative reactivities is that they clearly reveal small differences between reactivity scales. When discussing the influence of background conditions (lines 270 onwards), these would be much clearer if they were presented as ratios to ethene: $IRs/IR_{ethene}$. Table 2 should be replaced with Tables of $IR/IR_{ethene}$ values then we could see if the background conditions really changed the reactivities for particular VOCs. Also, in Figures 5 and 6, we see the differences in reactivity scales between Guangzhou and USA. But these would be much more illuminating if they were presented as $IR/IR_{ethene}$ values rather than as ranks. Ranks disguise the magnitudes of the differences.*

**Response:** thanks for the suggestion. We agree that ranks disguise the magnitudes of the

differences, and the results of ranks have been removed from the revised Figures 5 and 6.

We began with IRs and analyses were mainly done based on IRs, as most readers are familiar with the concept of IRs rather than RRs. For this purpose, we decided to save the results of IRs in the revised Figure 5. On the other hand, both the scenarios (U.S. vs. Guangzhou) and chemical mechanisms (SAPRC-07 vs. MCM v3.3.1) are different in the calculations of IR-Carter 2010 and IR-Guangzhou, so we cannot directly quantify the effects of background conditions by comparison between IR-Carter 2010 and IR-Guangzhou.

We used IR/Ethene to represent IRs/IR$_{ethene}$, and compared MIR/Ethene-CA-MCM with MIR/Ethene-GZ-MCM to determine the influence of environmental conditions (see below the revised Figure 6). With the same chemical mechanism (MCM), the MIR/Ethene scale resulted in higher consistency between Guangzhou and California ($R^2$ = 0.90) than did the U.S scenarios, but the MIR/Ethene magnitudes showed a large discrepancy (RMA slope: 1.54), implying the significant impact of environmental conditions on MIR/Ethene magnitudes.

The detailed information of California scenarios was not provided in Derwent et al. (2010), so we mainly examined the reasons for the discrepancy between IR-Carter 2010 and IR-Guangzhou. For this purpose, we decided to save the results of IRs in Table 2. Besides, we have added the results of RRs in the revised supplement (i.e., Table S5) to present the effects of environmental conditions on RRs. For clarity, we have added the following content in the revised manuscript.

"We also conducted similar analyses on RR scales, and the results confirmed that environmental conditions exert large impact on the RR magnitudes (Table S5)."

[Figure]

**Revised Figure 5**. Comparison of emission-based (a) MIR and (b) MOR scales for 111 common VOC species between Guangzhou and the U.S. The panels are in log scale, and only positively reactive VOCs are shown. The grey dashed line represents the 1:1 line. The

MIR-Carter 2010 and MOR-Carter 2010 data are taken from Carter et al. (2010).

[Figure]

1: *n*-nonane  2: *n*-decane  3: *n*-octane  4: benzene  5: styrene

**Revised Figure 6:** Comparison of the MIR/Ethene (the MIR value of a given VOC divided by the MIR value of ethene) values for 79 common VOC species between Guangzhou and California. Only positively reactive VOCs are shown, and the top five VOC species with a relatively large reactivity value change (shown below the panel) are marked with numbers. The grey dashed line represents the 1:1 line. The MIR-Ethene-CA-MCM data are taken from Derwent et al. (2010).

**Table S5a.** Dependence of emission–based MIR/Ethene scales for major VOC groups on major environmental conditions. The base MIR/Ethene scales were used as benchmarks (x-axis). Eight sensitivity scenarios: (2) 0.25*VOCs: both initial and emitted VOCs multiplied by 0.25, while other factors were kept unchanged, (3)-(6) X*VOC group: both initial and emitted VOC species contained in the target VOC group multiplied by X, and other VOC compositions increased or decreased to keep VOCs/NOx ratios constant, while other factors were kept unchanged, (7) HONO: both initial and emitted HONO were set to zero, and only the gas phase formation pathway (OH + NO = HONO) of HONO was included in the model, (8) 0.5*J: the photolysis rate was reduced to a half, while other factors were kept unchanged, (9) 3 days: the model integration time lasted for 3 days, while other factors were kept unchanged. Refer to Table S3 for the detailed speciation of VOCs.

| VOC group | 0.25*VOCs | | 0.25*OVOCs | | 1.5*Alkanes | | 2*Aromatics | | 2*Alkenes | | HONO | | 0.5*$J$ | | 3 days | |
|---|---|---|---|---|---|---|---|---|---|---|---|---|---|---|---|---|
| | RMA slope | $R^2$ | RMA slope | $R^2$ | RMA slope | $R^2$ | RMA slope | $R^2$ | RMA slope | $R^2$ | RMA slope | $R^2$ | RMA slope | $R^2$ | RMA slope | $R^2$ |
| Aldehydes | 0.90 | 0.99 | 1.12 | 1.00 | 1.21 | 1.00 | 1.07 | 1.00 | 0.89 | 1.00 | 1.10 | 1.00 | 1.62 | 0.98 | 0.73 | 0.83 |
| Alkanes | 1.04 | 0.99 | 1.01 | 1.00 | 1.06 | 0.99 | 0.99 | 1.00 | 0.98 | 1.00 | 1.00 | 1.00 | 1.17 | 0.92 | 1.11 | 0.43 |
| Alkenes | 0.77 | 0.97 | 1.14 | 1.00 | 1.16 | 1.00 | 1.19 | 0.99 | 0.86 | 0.99 | 1.14 | 1.00 | 1.64 | 0.97 | 0.46 | 0.64 |
| Aromatics | 1.02 | 1.00 | 1.03 | 1.00 | 1.07 | 1.00 | 0.93 | 1.00 | 0.97 | 1.00 | 1.00 | 1.00 | 1.21 | 0.98 | 0.73 | 0.95 |
| TVOC | 0.91 | 0.99 | 1.09 | 1.00 | 1.15 | 1.00 | 1.05 | 0.99 | 0.92 | 1.00 | 1.07 | 1.00 | 1.46 | 0.97 | 0.72 | 0.81 |

**Table S5b.** The same as Table S5a but for MOR/Ethene scales.

| VOC group | 0.25*VOCs | | 0.25*OVOCs | | 1.5*Alkanes | | 2*Aromatics | | 2*Alkenes | | HONO | | 0.5*$J$ | | 3 days | |
|---|---|---|---|---|---|---|---|---|---|---|---|---|---|---|---|---|
| | RMA slope | $R^2$ | RMA slope | $R^2$ | RMA slope | $R^2$ | RMA slope | $R^2$ | RMA slope | $R^2$ | RMA slope | $R^2$ | RMA slope | $R^2$ | RMA slope | $R^2$ |
| Aldehydes | 1.00 | 0.99 | 1.09 | 1.00 | 1.29 | 0.99 | 0.92 | 1.00 | 0.89 | 1.00 | 1.11 | 1.00 | 1.68 | 0.95 | 0.36 | 0.01 |
| Alkanes | 1.06 | 0.98 | 1.01 | 1.00 | 1.06 | 0.97 | 0.96 | 0.98 | 0.99 | 0.99 | 0.99 | 1.00 | 1.18 | 0.81 | 1.10 | 0.31 |
| Alkenes | 0.87 | 0.98 | 1.07 | 1.00 | 1.23 | 0.98 | 1.02 | 0.99 | 0.87 | 0.98 | 1.13 | 0.99 | 1.72 | 0.91 | 0.71 | 0.06 |
| Aromatics | 1.07 | 1.00 | 1.02 | 1.00 | 1.11 | 1.00 | 0.89 | 1.00 | 0.96 | 1.00 | 1.01 | 1.00 | 1.19 | 0.98 | 0.62 | 0.80 |
| TVOC | 0.99 | 0.99 | 1.06 | 1.00 | 1.21 | 0.99 | 0.94 | 1.00 | 0.91 | 1.00 | 1.08 | 1.00 | 1.50 | 0.97 | 0.48 | 0.42 |

*Specific Comments:*

*1. line 30: Agathokleous et al. is not the best reference to give here. There is an excellent reference available from the Tropospheric Ozone Assessment Report in Elementa.*

**Response:** the following three references have been cited in the revised manuscript.

"Fleming, Z. L., Doherty, R. M., von Schneidemesser, E., Malley, C. S., Cooper, O. R., Pinto, J. P., Colette, A., Xu, X., Simpson, D., Schultz, M. G., Lefohn, A. S., Hamad, S., Moolla, R., Solberg, S., and Feng, Z.: Tropospheric Ozone Assessment Report: Present-day ozone distribution and trends relevant to human health, Elem. Sci. Anth., 6, 10.1525/elementa.273, 2018."

"Lefohn, A. S., Malley, C. S., Smith, L., Wells, B., Hazucha, M., Simon, H., Naik, V., Mills, G., Schultz, M. G., Paoletti, E., De Marco, A., Xu, X., Zhang, L., Wang, T., Neufeld, H. S., Musselman, R. C., Tarasick, D., Brauer, M., Feng, Z., Tang, H., Kobayashi, K., Sicard, P., Solberg, S., and Gerosa, G.: Tropospheric ozone assessment report: Global ozone metrics for climate change, human health, and crop/ecosystem research, Elem. Sci. Anth., 6, 10.1525/elementa.279, 2018."

"Mills, G., Pleijel, H., Malley, C. S., Sinha, B., Cooper, O. R., Schultz, M. G., Neufeld, H. S., Simpson, D., Sharps, K., Feng, Z., Gerosa, G., Harmens, H., Kobayashi, K., Saxena, P., Paoletti, E., Sinha, V., and Xu, X.: Tropospheric Ozone Assessment Report: Present-day tropospheric ozone distribution and trends relevant to vegetation, Elem. Sci. Anth., 6, 10.1525/elementa.302, 2018."

*2. line 47: What is meant here? A mechanism is either explicit or not. If it contains non-stoichiometric chemical equations then it is not explicit.*

**Response:** sorry for the ambiguity. We have replaced "semi-explicit" with "detailed", as the SAPRC mechanism used for calculation of original IRs was more detailed than most of the "lumped" mechanisms.

*3. line 72: 'results obtained'.*

**Response:** changed.

*4. lines 86-87: It is widely recognised that the diurnal cycle in ozone is caused by changes in the stability of the boundary layer and not by intense in situ photochemical ozone production. This is explained in the Tropospheric Ozone Assessment Report.*

**Response:** the diurnal variations in surface ozone are mainly shaped by variations in photochemistry, boundary layer dynamics, surface dry deposition, and transport. The large amplitude of the diurnal $O_3$ cycle ($104 \pm 23$ ppbv) in urban Guangzhou is a combined result

of intense photochemical $O_3$ production during daytime and influences of NO titration and/or dry deposition on nighttime $O_3$ in the shallow nocturnal boundary layer. For clarity, we have made the following modifications in the revised manuscript.

"During the 67 $O_3$ episodes, the average amplitude (defined as the maximum minus the minimum) of the diurnal $O_3$ cycle was $104 \pm 23$ ppbv. The large amplitude indicated the significant influence of intense photochemical $O_3$ production during daytime as well as impact of NO titration and/or dry deposition during nighttime with shallow nocturnal boundary layer."

5. *line 93: ppbv ppbv$^{-1}$.*

**Response:** revised.

6. *line 99: Reference to the MCM website at the University of York would be more up-to-date.*

**Response:** changed.

7. *line 112: Where does these initial concentrations come from if not from observations? There are many sets of OVOC data for China.*

**Response:** observation data were available for 39 compounds. For compounds without available observation data (excluding formaldehyde and acetaldehyde), the initial concentrations of 0.10 ppbv were set according to the lowest observed concentration of all VOC species during the 67 selected $O_3$ episodes in Guangzhou. For formaldehyde and acetaldehyde, the initial concentrations of 0.50 and 0.30 ppbv were set according to model simulations with 1-day observation-based inputs. Such treatment in VOC initialization was used to represent scenarios with the lowest limit of VOC concentrations in Guangzhou.

We wondered the way to collect OVOC data from published studies, but it was difficult to get the daily input of OVOC concentrations, as most studies merely presented the average concentrations of VOCs.

We admit that such treatment in VOC initialization would cause uncertainty on the obtained RR scales. Sensitivity tests were conducted to evaluate the uncertainty, as documented in the revised Table S4 (see below). The initial concentrations of several selected compounds (either with relatively large or low IRs) were increased to high levels individually (ethanol: 10 ppbv; 1,3-dimethyl-5-ethyl: 2 ppbv; 2-methyl-2-butene: 1.5 ppbv; *i*-butene: 1.5 ppbv; methyl glyoxal: 1.5 ppbv), and such treatment would exert impact ranging from -14 % to 3 % on RR magnitudes, while $R^2$ fell in the range of 0.99-1.00. Besides, the initial concentrations of VOCs without available observation data were increased to 0.50 ppbv, and such treatment would exert impact of -12% and -11% on RR magnitudes under MIR and MOR scenarios, respectively, while $R^2$ equaled to 0.98 and 0.99, respectively.

We have clarified this uncertainty of the present study by adding the following statements in the revised manuscript.

"Such treatment of VOC initialization would inevitably cause uncertainty to the obtained RRs. Sensitivity tests were conducted to evaluate the potential uncertainty, and the results are documented in Table S4. The RRs (i.e., IR/Ethene in Table S4: IR value of a given VOC divided by the IR value of ethene) obtained from sensitivity tests exhibited good correlations with those obtained from emission-based base case inputs ($R^2$ ranged from 0.98-1.00), but minor discrepancy in RR magnitudes existed (reduced major axis (RMA) slope: 0.86-1.03). More high-quality long-term observational data covering a variety of VOCs are highly needed for IR calculation."

**Table S4.** Correlations of IR/Ethene scales under MIR and MOR scenarios for 116 VOC species between base case emission-based inputs and sensitivity scenarios. The base IR/Ethene scales were used as benchmarks ($x$-axis). Six sensitivity scenarios: (1) ADJ1: the initial concentration of ethanol increased to 10 ppbv, while other factors were kept unchanged; (2) ADJ2: the initial concentration of 1,3-dimethyl-5-ethyl increased to 2 ppbv, while other factors were kept unchanged; (3) ADJ3: the initial concentration of 2-methyl-2-butene increased to 1.5 ppbv, while other factors were kept unchanged; (4) ADJ4: the initial concentration of $i$-butene increased to 1.5 ppbv, while other factors were kept unchanged; (5) ADJ5: the initial concentration of methyl glyoxal increased to 1.5 ppbv, while other factors were kept unchanged; (6) ADJ6: the initial concentrations of VOCs without available observational data were set as 0.50 ppbv, while other factors were kept unchanged.

| MIR/Ethene | | | MOR/Ethene | | |
|---|---|---|---|---|---|
| Scenarios | RMA slope | $R^2$ | Scenarios | RMA slope | $R^2$ |
| ADJ1 | 1.03 | 1.00 | ADJ1 | 1.03 | 1.00 |
| ADJ2 | 0.95 | 1.00 | ADJ2 | 0.92 | 1.00 |
| ADJ3 | 0.89 | 0.99 | ADJ3 | 0.92 | 0.99 |
| ADJ4 | 0.96 | 1.00 | ADJ4 | 0.97 | 1.00 |
| ADJ5 | 0.86 | 0.99 | ADJ5 | 0.89 | 0.99 |
| ADJ6 | 0.88 | 0.98 | ADJ6 | 0.89 | 0.99 |

*8. line 128: Replace 'folds' with scaling factors.*

**Response:** done.

*9. line 134 onwards: please explain what the 'base' scenario is.*

**Response:** "base" scenario refers to the scenario whose NOx inputs were derived from observation or emission data, without any adjustment based on reactivity results. For clarity, we have made the following modifications in the revised manuscript.

"[NOx]$_{BASE}$ represents the NOx inputs under base scenarios that were directly derived from observation or emission data, without any adjustment based on reactivity results."

*10. line 176: The chemspider website reference provides rate coefficients presumably and not reaction fluxes.*

**Response:** sorry for the ambiguity. We have changed "fast reaction rates" to "large rate coefficients".

"This was reasonable considering their large rate coefficients of reactions with OH radicals (Calvert et al., 2015; McGillen et al., 2020)."

"Calvert, J. G., Orlando, J. J., Stockwell, W. R., Wallington, T. J.: The Mechanisms of Reactions Influencing Atmospheric Ozone, Oxford University Press, New York, 2015."

"McGillen, M. R., Carter, W. P. L., Mellouki, A., Orlando, J. J., Picquet-Varrault, B., and Wallington, T. J.: Database for the kinetics of the gas-phase atmospheric reactions of organic compounds, Earth Syst. Sci. Data, 12, 1203-1216, 10.5194/essd-12-1203-2020, 2020."

*11. line 238-242: It would be exceptionally useful if a little more detail was given here about why the five outliers are difficult to represent in chemical mechanisms. Benzene and styrene, presumably like phenol, are strong inhibitors of ozone formation. Is there a simple explanation how this works mechanistically. Then we have n-octane through n-decane. Presumably the mechanism of inhibition is different here and it would be interesting to know why this is. Why does it begin with n-octane and not n-heptane?*

**Response:** the major degradation pathways of the five outliers within MCM v3.3.1 have been present in Figure R1 to give a brief explanation. As shown, the major degradation products of benzene and styrene are phenol and benzaldehyde, respectively, which are strong inhibitors of $O_3$ formation under specific conditions.

For long-chain alkanes, the negative IR was primarily because of radical removal from reactions of peroxy radicals with NO forming organic nitrates. Inhibition due to nitrate formation was not begun with n-octane but was increasingly important as the size of the molecule increases. Overall, the IRs for such compounds reflect a balance between the positive effects on $O_3$ due to the conversions of NO to $NO_2$ by the radicals they form, and also due to the reactivity of its products, and the negative effects caused by radical inhibition. Both effects are large and their magnitudes are sensitive to scenario conditions and to the base mechanism, as well as to details on how the reactions of the compound and its product are represented in the mechanism. For that reason, the overall effect, which is the difference between two large and variable numbers, can vary from positive to negative, depending on the scenario and mechanism used.

For clarity, we have made the following modifications in the revised manuscript.

"The IRs for the former three long-chain alkanes reflect a balance between the positive effects on $O_3$ due to the conversions of NO to $NO_2$ by the radicals they form, and also due to

the reactivity of its products, and the negative effects caused by radical inhibition (organic nitrates). This is also the case for benzene and styrene, whose major degradation products are phenol and benzaldehyde, respectively, which are strong inhibitors of $O_3$ formation under specific conditions."

**Figure R1a.** The major degradation pathways of benzene (BENZENE) to form phenol (PHENOL).

**Figure R1b.** The major degradation pathways of styrene (STYRENE) to form benzaldehyde (BENZAL).

**Figure R1c.** The major degradation pathways of n-octane (NC8H18) to form octan-3-yl nitrate (OCTNO3).

**Figure R1d.** The major degradation pathways of n-nonane (NC9H20) to form nonan-3-yl nitrate (NONNO3).

**Figure R1e.** The major degradation pathways of n-decane (NC10H22) to form decan-3-yl nitrate (represented as DECNO3).

*12. line 241: This point would be self-evident if the presentation had been done with MIR/MIR$_{ethene}$ ratios.*

**Response:** the analyses mentioned were actually done based on MIR/MIR$_{ethene}$ ratios (which were represented as "MIR/Ethene" in the revised Figure 6 and Section 3.2). The details have been provided in response to "General Comments".

*13. line 244: MIR*

**Response:** "MR" is the abbreviation of mechanistic reactivity mentioned in line 41 in the revised manuscript. For clarity, "MR" has been changed to "reaction mechanisms".

*14. line 250: What was done with one quarter of the base ratios?*

**Response:** "one quarter of the base ratios" refer to the sensitivity scenario of "0.25*VOCs" in Table 2, in which both initial and emitted VOCs multiplied by 0.25, while other factors were kept unchanged. For clarity, we have made the following modifications in the revised manuscript.

"One quarter of the base VOC/NOx ratios (both initial and emitted VOCs multiplied by 0.25, while other factors were kept unchanged) in the Guangzhou scenarios tended to increase the overall IR values (by 19 % and 26 % under the MIR and MOR scenarios, respectively)."

*15. Section 3.2: This is a big section that would benefit considerably from being split into smaller sub-sections.*

**Response:** the original Section 3.2 has been divided into two sections, with the first being

"3.2 Comparison with ozone reactivity scales for VOCs in U.S.", and the second being "3.3 Other possible reasons for the IR discrepancy between China and the U.S. scenarios".

*16. line 336 onwards. This is an important recommendation and should be in the Conclusions section with a little more explanation.*

**Response:** we agree with this, and have added the following descriptions in "Conclusions" of the revised manuscript.

"Considering the huge impact of chemical mechanisms and environmental conditions on the IR scales, more systematic comparisons focusing on VOC reactivity scales obtained from the U.S. vs. Chinese conditions using the same and different chemical mechanisms are needed."

**Response to Anonymous Referee #2**

*General Comments:*

*In this paper, Carter's method and local data of Guangzhou were applied to construct new MIR, MOR and EBIR scenarios under observation and emission methods through a box model equipped with MCMv3.3.1 mechanism. At the same time, the MIR-characteristics of VOCs are described and analyzed. The application of recent research results to achieve migration is meaningful and has universality in application. At the same time, the research needs to carry out data processing for each VOC, and the workload is heavy, which is worthy of recognition. After review, it is considered that the article still needs to address the following concerns:*

**Response:** we thank the reviewer for the positive comments and constructive suggestions, which are much helpful for improving the original manuscript. We have carefully considered all of these comments and revised the manuscript accordingly. Below we provide the original reviewer's comments in black italic, with our response and changes in the manuscript in blue and red, respectively.

*Specific Comments:*

*1. For the box model, the observation based method adopted by the author to constrain the concentration of both $NO_2$ and NO at the same time, which made the $O_3$ concentration largely fixed by the ratio of $NO_2/NO$. In this case, could the impact of VOCs on $O_3$ be reasonably reflected?*

**Response:** thanks for the helpful comment. For the present modelling, at the beginning of each integration step (1 h), the NO and $NO_2$ inputs into the model were determined by observational data with observation-based inputs, but their evolution over time was determined by the chemistry that was affected by the VOC-involved reactions. For charity, we have added the following descriptions in the revised manuscript.

"While the NO and $NO_2$ inputs into the model were determined by observational data with observation-based inputs, their evolution over time was determined by the chemistry that was affected by the VOC-involved reactions"

*2. The running step of the box model is 1h, is it too sparse for the total integral period of 10h? In the paper, it may be necessary to include the graphs or tables of $O_3$ concentration changes in the two observation-based and emission models within 10 hours.*

**Response:** we have calculated the RRs with running step of 10 min for comparison. With observation-based inputs, the CO, $SO_2$, NO, $NO_2$, HONO, and VOCs concentrations were averaged or interpolated into a 10-min time resolution to constrain the model. With

emission-based inputs, the emission rates (unit: molecules $cm^{-3}$ $s^{-1}$) of NO, $NO_2$, VOCs (116 compounds), $SO_2$, and CO were read in with a time resolution with 10 min. Besides, meteorological parameters, including temperature, RH, and pressure were averaged or interpolated into a 10-min time resolution to constrain the model with both inputs.

As Figure R1 shows, the RRs obtained from 10-min inputs exhibited good correlations with those obtained from 1-h inputs ($R^2$: 0.98-1.00). The discrepancy in RRs magnitudes existed but was relatively small, especially for emission-based inputs (RMA slope: 1.02 under MIR scenarios and 1.03 under MOR scenarios). Considering the heavy computation burden introduced by 10-min integration step, step of 1-h is a good choice for RR calculations.

The graphs of $O_3$ concentration with 1-h observation-based and emission-based inputs have been included below as well as in the revised SI (Figure S3).

[Figure]

**Figure R1**. Comparison of RRs for VOCs obtained from 1-h and 10-min inputs. The left panel is for emission-based inputs, and the right panel is for observation-based inputs. The RMA represents reduced major axis. The black dashed line represents the 1:1 line.

[Figure]

**Figure S3**. The model-simulated $O_3$ concentrations with emission-based and

observation-based inputs during the 10-h integral period.

*3. The MIR table of VOCs species is suggested to refer to the article published by Carter in 2007, which is arranged in the order of commonly used alkanes, olefins and aromatic hydrocarbons.*

**Response:** thanks for the suggestion. We have re-arranged the order of VOC species in IR table as follows.

[revised manuscript text omitted]

*4. Fig. 5a is one of the most important conclusions of the whole paper, which is used to compare the MIR-values of the article and the MIR-values in the literature. Compared with the logarithmic axis, the comparison results of the conventional axis are more convincing. At the same time, scatter plots similar to the size order in Fig. 5b should be reduced, because the deviation of MIR order in a considerable number of VOCs species is large, and R² is of little significance. This paper needs to further prove the validity of the calculated MIR in Guangzhou, so it can be used to replace the MIR from US (mainly Carter's publication) for Guangzhou.*

**Response:** thanks for the suggestion, but the logarithmic scale may actually be more useful for displaying differences in reactivity results, so the readers can see the data reasonably well at any magnitude range. In addition, the results of ranks have been removed in the revised

Figures 5 and 6.

We adopted and improved Carter's method for MCMv3.3.1 to calculate IRs under MIR, MOR and EBIR scenarios in Guangzhou, and the IR-Guangzhou were compared with those provided in Carter (2007). A nonnegligible discrepancy was found between IR-Guangzhou and IR-Carter, and our results revealed the significant impact of environmental conditions on IR/RR magnitudes. Rough application of U.S.'s IR scales in China may ambiguously identify the reactivity of VOCs, especially those relied heavily on chemical conditions such as styrene and long-chain alkanes. Besides, the discrepancy in IR magnitudes would introduce uncertainty to the OFP quantification. Take the above factors into account, it would be better to use localized IRs to provide scientific support for VOCs control in Guangzhou.

[Figure]

**Revised Figure 5**. Comparison of emission-based (a) MIR and (b) MOR scales for 111 common VOC species between Guangzhou and the U.S. The panels are in log scale, and only positively reactive VOCs are shown. The grey dashed line represents the 1:1 line. The MIR-Carter 2010 and MOR-Carter 2010 data are taken from Carter et al. (2010).

[Figure]

1: *n*-nonane  2: *n*-decane  3: *n*-octane  4: benzene  5: styrene

**Revised Figure 6:** Comparison of the MIR/Ethene (the MIR value of a given VOC divided by the MIR value of ethene) values for 79 common VOC species between Guangzhou and California. Only positively reactive VOCs are shown, and the top five VOC species with a relatively large reactivity value change (shown below the panel) are marked with numbers. The grey dashed line represents the 1:1 line. The MIR-Ethene-CA-MCM data are taken from Derwent et al. (2010).

*5. In Table 2a, why change the simulation time to 3 days? 10 h can describe the period of time during which VOCs receive light and undergo photolysis reaction cycle in real day, and the situation of 3 days lacks practical significance.*

**Response:** the 3-day simulation was used to characterize a regional scenario which extended the IR evaluation from urban scale to regional scale, as the $O_3$ pollution is a regional environmental problem and the relative importance of VOCs and NOx as well as of different VOC species to $O_3$ formation may be different between urban and regional scales.

For clarity, the following modifications have been made in the revised manuscript.

"A regional scenario (the model integration time lasted for 3 days, whereas the other factors were kept unchanged; indicated as "3 days" in Table 2) was designed to evaluate the relative importance of VOCs and NOx as well as of different VOC species to $O_3$ formation in regional scales."

*However, for the MIR scenario and MOR scenario, there were differences in RMA slope changes in the 3-day simulation, and some $R^2$ were too small. It needs to be clarified.*

**Response:** Because NOx is consumed and removed rapidly, the NOx emissions rather than VOC emissions from upwind urban sources would play more important roles in $O_3$ concentrations over large regional scales. This is confirmed by the overall lower MIRs and MORs in the 3-day scenarios than in the 10-hour scenarios. The NOx conditions under MIR scenarios are higher than under MOR scenarios, therefore, the MORs dropped much faster than MIRs in 3-day scenarios.

The reactive compounds (such as alkenes) typically react near its source and contribute to the photochemical production of $O_3$ in the area in which it is emitted, but the fast consumption made their IRs show large downward trends in 3-day regional scenarios. In contrast, the unreactive compounds (such as alkanes) have a longer residence time in the atmosphere, and they would build up and undergo extensive photochemical reactions in 3-day regional scenarios. Besides, the unreactive compounds and their oxidation products would indirectly impact the IRs of other compounds by exerting effects on radical recycling. Taken together, the unreactive compounds played relatively more important roles in 3-day regional scenarios and their IRs showed slowly downward or even upward trends compared to those calculated

based on 10-hour scenarios. The opposite trends explained why $R^2$ were relatively small for alkanes and alkenes between 3-day and 10-hour scenarios. This also explained the poor correlations in IRs for aldehydes between 3-day and 10-hour scenarios, as aldehydes' IRs showed a wide distribution and showed varying sensitivity to the changes of NOx availability.

For clarity, the following modifications have been made in the revised manuscript.

"The overall MIR and MOR values for the 116 VOCs changed by -5 % ($p$ = 0.07) and -61 % ($p$ < 0.01), respectively, implying that the NOx emissions rather than VOC emissions from upwind urban sources would play more important roles in $O_3$ concentrations over large regional scales. The NOx conditions under MIR scenarios are higher than under MOR scenarios, therefore, the MORs dropped much faster than MIRs in 3-day regional scenarios."

"Along the 3-day scale, the reactive VOC groups (such as alkenes) with a short lifetime were rapidly consumed in urban scales, but their IRs showed fast downward trends in 3-day regional scales (e.g., the median rank of alkenes dropped by -11 and -27 under the MIR and MOR scenarios, respectively). In contrast, the role of unreactive VOC groups became more important, which would build up and undergo extensive photochemical reactions along the 3-day scale (Stockwell et al., 2001). Besides, the unreactive VOC groups (such as alkanes) and their oxidation products would indirectly impact the IRs of other compounds by exerting effects on radical recycling. Taken together, the IRs for unreactive VOC groups showed slowly downward or even upward trends in 3-day scenarios than 10-hour scenarios. The opposite trends explained why $R^2$ were relatively small for some VOC groups between 3-day and 10-hour scenarios."